# Understanding Long Noncoding RNA and Chromatin Interactions: What We Know So Far

**DOI:** 10.3390/ncrna5040054

**Published:** 2019-12-03

**Authors:** Kankadeb Mishra, Chandrasekhar Kanduri

**Affiliations:** 1Department of Medical Biochemistry and Cell Biology, Institute of Biomedicine, University of Gothenburg, 40530 Gothenburg, Sweden; kankadeb@gmail.com; 2Department of Cell Biology, Memorial Sloan Kettering Cancer Centre, Rockefeller Research Laboratory, 430 East 67th Street, RRL 445, New York, NY 10065, USA

**Keywords:** lncrna, chromatin, chromatin RNA, long noncoding RNA, gene regulation, RNA-chromatin interactions

## Abstract

With the evolution of technologies that deal with global detection of RNAs to probing of lncRNA-chromatin interactions and lncRNA-chromatin structure regulation, we have been updated with a comprehensive repertoire of chromatin interacting lncRNAs, their genome-wide chromatin binding regions and mode of action. Evidence from these new technologies emphasize that chromatin targeting of lncRNAs is a prominent mechanism and that these chromatin targeted lncRNAs exert their functionality by fine tuning chromatin architecture resulting in an altered transcriptional readout. Currently, there are no unifying principles that define chromatin association of lncRNAs, however, evidence from a few chromatin-associated lncRNAs show presence of a short common sequence for chromatin targeting. In this article, we review how technological advancements contributed in characterizing chromatin associated lncRNAs, and discuss the potential mechanisms by which chromatin associated lncRNAs execute their functions.

RNA, considered a potential primordial molecule of life, has been serving as a central molecule in molecular biology research due to its ability store information and execute catalytic functions in diverse biological contexts. It has long been argued that RNA mediated informational activities have slowly evolved into a more stable and easily replicable DNA, while the catalytic functions have evolved into highly versatile polypeptides [1]. However, experiments by Beadle and Tatum led to the proposition of “one gene-one enzyme” hypothesis (simplified to a more popular “one gene-one protein” assumption) [2] which formed the basis for the formulation of central dogma of molecular biology [3,4]. This concept that genes solely encoded the functional components of cells in the form of proteins (i.e., the ‘enzymes’) was influenced by the limitations of experimental and technological advancements in the field that shaped the perception of molecular mechanism of this era. This theory minimized the functional potential of RNA as a molecule merely bridging DNA to protein. Accordingly, an increase in the number of protein coding genes was proposed to correlate positively with the increase in organismal and molecular complexity. However, the most convincing evidence against this theory came with the advancement of technology that allowed generation of high throughput transcriptome sequencing data from several model organisms across the evolutionary ladder. These data proved beyond doubt that the number of protein coding genes does not correlate with the organismal complexity [5]. However, while the genomes of higher eukaryotes pervasively transcribe to form RNA, only a small percentage of the transcribed RNAs were found to be translated into proteins [6]. This noncoding portion of the transcribing genome (termed non-coding RNA) has consistently increased with organismal complexity [7]. Consistent with the latter notion, recent evidence demonstrated that long noncoding RNAs play a critical role in tissue and developmental-dependent biological functions, suggesting that noncoding portion of the genome plays a significant role in rewiring multi-layered gene expression controlling the organismal development.

Noncoding RNAs (ncRNAs) are broadly classified into two categories based on their functions i.e., those involved in housekeeping function (tRNA, rRNA) and the others which are regulatory in nature. Regulatory ncRNAs are further classified based on their size as small ncRNA (microRNA, siRNA and piRNA), short ncRNAs (snoRNA and snRNA) and long ncRNA (lncRNA). LncRNAs are arbitrarily defined as transcripts of 200 nucleotides (nt) or more and lacking an open reading frame (ORF) (typically have an ORF of not more than 30 amino acids) [8]. Taken together, ncRNAs generate a complex transcriptional output in mammals in addition to the limited number of protein coding genes. Based on their position relative to the neighboring protein coding genes, lncRNAs can be classified into two sub-classes: sense lncRNA and an antisense lncRNA. Further, based on diversity in their biogenesis, distribution, structural, functional, and mechanism of action has resulted in several ways of sub classifying regulatory lncRNAs in an effort to consolidate for unifying features of any sub-class. Based on such efforts, lncRNAs can be broadly categorized into the following ways of subclassification:**Loci of biogenesis**: Intergenic lncRNAs (lincRNAs), intronic lncRNAs, natural antisense transcripts (NATs), bidirectional-promoter lncRNAs, enhancer RNAs (eRNAs) [9,10,11], promoter associated RNAs (PARs) [5,12,13,14] and terminus associated RNAs (TARs) [14,15,16].**Distribution**: Nuclear lncRNAs, cytoplasmic lncRNAs [17,18,19,20,21] and mitochondrial lncRNAs [22,23].**Structure**: Linear lncRNAs and circular lncRNAs (circRNAs) [24,25,26,27].**Site of action**: *Cis*-acting lncRNAs [28,29,30,31,32], *trans*-acting lncRNAs [33,34] and competing endogenous RNAs (ceRNAs) [35,36,37,38].**Mechanism of action**: Acting as scaffold, decoy, guide and signal [39,40,41].

The above classifications of lncRNAs represent one of the most simplistic approaches of sub classification and might not include all the categories. Moreover, the basis of many of these classifications can have overlapping implications, for example, a linear lncRNA that is localized in the nucleus act as a scaffold in recruiting chromatin modifying complexes for the regulation of its target genes *in cis* (Table 1). Several lncRNAs are also known to exhibit pleiotropic effects [20,42] making such categorizations to be error prone.

It has recently been shown that several lncRNAs regulate gene expression in critical cellular contexts by organizing chromatin into active and inactive domains through direct interaction with different chromatin modifying enzymes [28,33,57]. For example, sense (*Xist*) and antisense (*Tsix*) pair of lncRNAs from the mammalian X chromosome inactivation center (XIC) exclusively work *in cis*-acting fashion to regulate chromatin structure to bring about X chromosome inactivation. *Xist* and *Tsix* lncRNAs have been acting as paradigms for chromatin dependent gene regulation by lncRNAs on whole chromosome and at a single gene level, respectively. In particular, *Xist*, by scaffolding several chromatin modifying enzymes, it has been shown to regulate higher order (3D) chromatin structure of the X-chromosome during the onset of X chromosome inactivation (XCI) [31,58,59]. Likewise, several antisense lncRNAs from the imprinted clusters have also been shown to regulate gene expression in large chromosome domains through organizing higher order chromatin structure (Table 1). Although we begin to understand the functional role of lncRNAs and their interacting proteins in the regulation of chromatin structure, but how they are targeted to chromatin is not very well understood. In this review, we will focus on the regulatory functions of chromatin associated lncRNAs and how the emergence of various technological advances has enabled comprehensive annotation and functional identification of these transcripts. Finally, we will discuss the mechanisms by which chromatin associated lncRNAs make contacts with their target genes.

## 1. Approaches to Define RNA-Chromatin Interactions

In the early 1980s, elegant genetic and molecular studies identified a phenomenon of parent-of-origin-specific allelic expression called genomic imprinting which explained in part the molecular mechanism of dosage compensation [60]. Independently, two imprinted genes were identified: the paternally expressed protein-coding gene *Igf2* and the maternally expressed lncRNA *H19*. Both genes map to the distal end of the mouse chromosome 7, which lie in proximity to each other forming the *H19/IGF2* cluster [61,62]. *H19* lncRNA was among the first lncRNAs to be functionally characterized in various biological contexts, including genomic imprinting. This era witnessed a burst in the identification and functional characterization of several imprinted lncRNAs with chromatin regulatory functions, resulting in a gradual explosion of different techniques over the next decade to address the mechanism of lncRNA interaction with chromatin modifiers or other proteins and with chromatin (Figure 1). One of the most defining experimental evidence, implicating lncRNAs in chromatin organization, came in 1991 when *Xist* lncRNA was shown to localize to the inactivated X chromosome [63]. This observation was followed up by several other studies where imprinted lncRNAs were all found to execute their actions by being in close interaction with chromatin [42,64]. Mechanistic studies of imprinted lncRNAs for their role in the regulation of imprinted gene clusters were based on experimental approaches that were locus- or gene specific, where localization and binding protein partners were identified for any given lncRNA (Table 1). These mechanistic studies based on imprinted lncRNAs, inspired to develop experimental approaches that can identify lncRNAs which can bind to a given protein, in particular to different chromatin modifiers such as PRC2 [65,66,67], YY1 [68,69], CTCF [70] and others [71]. These “protein centric” approaches (also refer Box 1) led to the global identification of lncRNAs that bind to several chromatin modifiers and thus possibly to chromatin. These approaches have identified potential chromatin interacting lncRNAs, however with few exceptions, direct targeting of lncRNAs to chromatin was not validated. This led to the next wave of experimental approaches with a focus to identify more direct evidence for lncRNA-chromatin interactions. These approaches can be broadly divided as RNA and non-RNA centric approaches as discussed below.

Box 1Methods to study global RNA-protein interactions.***RIP-seq*:** RNA immunoprecipitation (RIP) exploits antibodies to pull down RNA bound to a given protein and the immunoprecipitated RNA subjected to high throughput sequencing (RIP-seq), thereby, enabling global identification of RNAs bound to protein of interest. Technical variants of this methods include native RIP-seq [72] and formaldehyde cross linked fRIP-seq [71].***RIPiT-Seq*:** RNA: Protein immunoprecipitation in tandem (RIPiT) is suitable for RBPs with poor inherent ultraviolet (UV) crosslink ability. This method yields highly specific RNA binding footprints of any cellular RNPs and the resulted RNA footprints can then be combined with high-throughput sequencing (RIPiT-Seq) thereby providing a means to map the RNA binding sites of such RBPs [73]. This method has been used to identify and validate RNA binding pocket within WDR5 chromatin modifier [74].***CLIP*:** Improves the specificity of RIP by UV crosslinking of RNA/protein complexes before extraction. This allows the removal of weakly bound RNA through stringent washing. The remaining RNA can then undergo reverse transcription and PCR amplification (or next generation sequencing). The main drawback of this method is the loss of a significant proportion of transcripts which are stalled at the cross-linking site resulting in truncated cDNAs. UV crosslinking can also introduce some bias as its ability to bind RNA to protein varies depending on the base/proximity of the reactive amino acids mediating the interaction. ***HITS-CLIP*** when CLIP is combined with high throughput next generation sequencing [75].***iCLIP*:** Individual-nucleotide-resolution CLIP (iCLIP) was developed to enable the recovery of truncated cDNAs lost in conventional CLIP. The iCLIP protocol employs UV irradiation as a cross-linking source that preserves in vivo RNA-protein interactions through promoting covalent bonds at the sites of protein-RNA interactions. Following mild RNAse treatment, to obtain RNA fragments in an optimal size range, RNA-protein complexes are immunoprecipitated. The immunoprecipitated RNA is dephosphorylated to enable an adapter ligation to the 3′ end of the RNA and radioactive labelling at the 5′ end. This method includes SDS-PAGE separation and transfer to nitrocellulose membrane to capture radiolabelled, immunoprecipitated, crosslinked RNA-protein complexes. The captured RNA is then reverse transcribed into cDNA. Following cDNA circularization, restriction enzyme digestion to linearize the cDNA prior to PCR amplification and library preparation for high-throughput sequencing. Truncated cDNA represents the majority in the cDNA library and the position of the preceding nucleotide, after mapping to the genome, corresponds to the cross-linking site (Huppertz et al., 2014).***eCLIP*:** Enhanced CLIP improves library preparation and circular ligation steps of iCLIP allowing greater power in filtering and mapping truncated sequences. eCLIP replaces the 5′ adaptor ligation with a 3′ cDNA ligation [76], whereas further improved eCLIP protocol Monitored eCLIP (meCLIP) uses both a 5′ ligation and a 3′ cDNA ligation [77].***irCLIP*:** This is similar to iCLIP apart from the fact that it makes use of a biotinylated, fluorescent 3′ DNA adaptor [78].***BrdU-CLIP*:** BrdU-CLIP built on the same principle as that of CLIP and iCLIP but employs a nucleotide analogue BrdUTP in reverse transcription to capture truncated and non-truncated cDNA products using BrdU antibody [79].***GoldCLIP*:** Improved with a shortened iCLIP protocol that removes the SDS-PAGE separation and membrane transfer steps. RNPs are tagged with Halo-tag and overexpressed in cell line of interest. Halo-tagged protein-RNA complex affinity purified using Halo-ligand. Following denaturing washes, the purified RNAs subjected to high-throughput sequencing [80].***PAR-CLIP*:** Photo-Activatable Ribonucleoside enhanced Cross-Linking and Immunoprecipitation (PAR-CLIP) is a modified CLIP method, where the introduction of photo-activated nucleosides in the media are taken up by cells and subsequently used for protein–RNA crosslinking thereby enabling the following advantages. First, PAR-CLIP shows in general 100- to 1,000-fold higher RNA recovery, in comparison to the conventional cross-linking at 254 nm. Secondly, UV radiation-induced T-to-C mutations characteristic of the cross-linked sites that have incorporated photo-activated nucleoside analogues. Based on this, PAR-CLIP exploits mutation analysis to improve the identification of precise RBP binding positions or footprint [81]. Studies using PAR-CLIP have identified that both EZH2 and JARID2 can directly interact with RNA in cells [82,83]. Their interaction with RNA is mutually exclusive and antagonistic to their ability to interact and bind to chromatin.***fCLIP*:** Formaldehyde cross-linking, immunoprecipitation and sequencing (fCLIP) uses formaldehyde as a cross-linking reagent for CLIP to characterize the RNA binding protein binding regions on double stranded RNAs. dsRNAs are inefficiently crosslinked by UV, thus making it difficult to study the interactions between dsRNA binding proteins and their substrates. It has been used to characterize mapping of in vivo DROSHA cleavage sites at single nucleotide resolution [84].***mRNA Interactome Capture (MIC)*:***mRNA Interactome Capture* (MIC) is an oligo dT-based capture of global mRNA protein-interactome from cells cross linked with complementary crosslinking chemistries: with UV (at 254 nm) or photoactivatable-ribonucleoside (4SU, 4 thiouridine)-enhanced crosslinking (PAR-CL) at 265 nm. This method characterized global mRNA proteome comprising novel RNA binding proteins, including metabolic enzymes. These two complementary chemistries allow a comparative analysis of the enriched RBPs. This investigation highlights the presence of intrinsically disordered structures in the large portion of the human proteome [85].***RBDmap*:*****RNA*** Binding Domain map *(RBDmap)* is an improved protocol of RIC, which finemaps the protein domains that interacts with mRNAs. UV irradiated cells were given stringent denaturing washes to purify the resulting covalently linked RBP–RNA complexes with oligo(dT) magnetic beads. As a defining modification to RIC, post elution the RBPs were subjected to partial proteolysis to retain only those protein regions that are bound to the RNA and are separated by a second oligo(dT) selection from the non-interacting peptides that are released into the supernatant. Mass-spectrometric analysis of the eluted and released peptides to calculate peptide intensity ratios between these fractions will determine the RNA-binding regions [86].***OOPS*:** Orthogonal Organic Phase Separation (OOPS) is a method based on UV cross linking of cells at 254 nm followed by Acidic Guanidinium Thiocyanate-Phenol-Chloroform (AGPC) phase separation, where RNA and proteins fractionated into the upper aqueous phase and the lower organic phase, respectively. Whereas RNA-protein adducts, generated by UV crosslinking, separated into the aqueous-organic interface. This interface accumulated RNA-protein adducts at the interface represent reliable RNA binding proteins on global scale which are in specific interaction with RNA [87].

## 2. RNA Centric Methods to Study Global RNA-Chromatin Interactions

### 2.1. Chromatin Oligoaffinity Precipitation (ChOP)

ChOP is one of the first techniques developed to affinity purify chromatin associated lncRNAs using biotinylated oligonucleotides. This technique was first used to determine the occupancy of Alu and B2 RNAs at the promoters of the repressed genes during heat shock [88]. Later this technique was successfully employed to characterize the occupancy of imprinted lncRNA *Kcnq1ot1* across the 1 Mb *Kcnq1* imprinted cluster [32]. This technique set a stage for the development of several other techniques based on the usage of biotinylated oligonucleotides on cross-linked chromatin to characterize interacting lncRNA binding proteins, and also genome-wide binding sites for lncRNAs.

### 2.2. RNA Antisense Purification (RAP)

This technique captures a target RNA of interest through hybridization with antisense biotinylated oligos. By cross-linking endogenous macromolecular complexes prior to RNA capture with different cross-linking agents such as 4′-aminomethyltrioxalen (AMT), a psoralen-derivative crosslinker, formaldehyde and disuccinimidyl glutarate (DSG), RAP allows for the identification of RNA, proteins and DNA loci that cross-link to and co-purify with the target RNA. While RAP-RNA approach elucidated the functions of *U1* small nuclear RNA and *Malat1* in RNA processing, the RAP-DNA technique enables genome-wide mapping of RNA-DNA interactions. Thus, RAP provides an important tool for systematic interrogation of lncRNA function and mechanism. RAP was used to study the localization of *Xist* during the onset of XCI. *Xist* lncRNA exploits the three-dimensional genome architecture to spread along the inactive X chromosome [31]. This also helped in dissecting the mechanism of *Firre* lncRNA to spatially assess inter chromosomal interactions [89,90].

### 2.3. Chromatin Isolation by RNA Purification (ChiRP)

Tilling antisense biotinylated oligonucleotides were used to retrieve specific lncRNA of interest bound to chromatin and proteins, which can be assayed separately, thus, generating a complete interaction profile for that particular lncRNA. Drosophila *rox2* lncRNA, human telomerase RNA *TERC* and *HOTAIR* lncRNAs were found to have precise but distinctly different *trans*- genomic targets [34]. Like ChOP, this technique enabled the identification of both *cis* and *trans* genomic targets of lncRNAs along with the identification of their interacting protein partners. The specificity and robustness of this technique enabled efficient interrogation of lncRNA functions.

### 2.4. Capture Hybridization Analysis of RNA Targets (CHART)

Like ChOP, ChIRP and RAP-DNA, CHART was developed to map the genome-wide binding profile of the chromatin-associated RNAs. CHART, ChOP, ChIRP and RAP-DNA built on the same principle i.e., purifying RNA associated chromatin regions using biotinylated antisense oligonucleotides. This protocol also uses a small number of 22–28 nucleotide antisense oligonucleotides, complementary to single stranded regions of a target RNA that are accessible for hybridization, to purify RNAs from a cross-linked chromatin extract. RNA–chromatin complexes immobilized on beads are eluted using RNase H, and the eluted genomic DNA is subsequently sequenced using high-throughput sequencing technologies and mapped to the reference genome to identify RNA–chromatin associations. The technique was initially used to successfully determine the genome-wide binding profile of the *roX2* ncRNA, a regulator of dosage compensation, in *Drosophila* S2 cells. This technique was later applied to map genome-wide chromatin occupancy for several mammalian long ncRNAs, including *Xist, Neat1* and *Malat-1* long ncRNAs [91,92,93].

The key differences between CHART and other three techniques i.e., RAP, ChOP and ChIRP are as follows:CHART uses a two-step formaldehyde cross-linking approach to fix nuclei.RNase H sensitivity assay is used to identify regions in the target RNA that are accessible for hybridization with antisense oligonucleotides. A small number of short oligonucleotides that have been predetermined to interact with the RNA target are then used in CHART to enrich for RNA–chromatin complexes.Antisense oligonucleotide bound RNA–chromatin complexes are eluted using RNase H. This reduces nonspecific false positive binding events generated by direct binding of antisense oligonucleotide probes to DNA [92].

The overall limitation of these oligonucleotide-based approaches is that the efficacy of these methods is based on the size of the lncRNA in question. Cellular localization of lncRNAs also affect the efficiency of this method. Transcript abundance along with *cis-* versus *trans-* action can also lead to inherent differences in the retrieval efficiencies of purified RNAs. These issues would limit investigations to specific lncRNAs and thus there is a need for alternate approaches that overcome above limitations in part. The need for such an approach possibly led to the gradual development of several techniques for identification of global targeting of RNAs to chromatin as we shall discuss next.

## 3. Non-RNA Centric Methods to Study Global RNA-Chromatin Interactions

### 3.1. Chromatin RNA Isolation by Sucrose Gradient Fractionation

In one of the initial efforts to globally characterize ncRNAs associated with chromatin, human skin fibroblast cells (HF) were treated with micrococcal nuclease (MNase) followed by separation of different length chromatin fragments on a sucrose gradient. Soluble chromatin fraction was collected from the gradient, and RNA was isolated and subjected to high throughput sequencing [57]. This effort led to the initial characterization of several evolutionarily conserved chromatin associated RNAs (CARs) which mostly mapped to intronic genomic regions. One of the intergenic CARs namely, *Intergenic10* was functionally characterized and it has been shown to positively regulate the transcription of non-overlapping nearby genes *in cis*. Importantly, ncRNA field, then, was conceptually predominated with studies understanding the functional significance of ncRNAs in the regulation of genomic imprinting [32,45,67]. Interestingly, in all these contexts, ncRNAs were regulating imprinted status by transcriptional repression of target genes. Thus, positive transcriptional regulation by *Intergenic10* CAR of neighbouring genes also opened up the prospect of other ncRNA mediated and/or assisted gene activation mechanisms.

### 3.2. Chromatin RNA Immunoprecipitation (ChRIP)

Pre high-throughput sequencing era witnessed robust application of this technique to validate the enrichment of lncRNAs in different chromatin compartments using antibodies against different histone modifications to pull down soluble chromatin fraction. RT-qPCR with specific primers were used to validate candidate lncRNA enrichment in the pull-down chromatin fractions. Using this technique, both human and mouse *Kcnq1ot1* lncRNAs were found to be enriched in repressive chromatin fractions [29,30].

With advancement in RNA sequencing technologies, and improved efficiency of library preparation methods, it was possible to combine RNA sequencing with ChRIP pull down RNAs (ChRIP-seq). Eventually, this procedure was modified by combining photoactivatable ribonucleoside-enhanced crosslinking followed by high-throughput RNA-sequencing to characterize lncRNAs that are associated with different chromatin fractions [94]. Using this modified ChRIP assay, repressive chromatin was purified using antibodies against H3K27me3 and EZH2, and sequencing of RNA purified from these fractions identified 276 commonly enriched chromatin-associated lncRNAs. EZH2, a component of the PRC2 complex, catalyses trimethylation of lysine 27 of histone H3 (H3K27me3), a repressive histone modification associated with gene silencing [33]. Thus, the lncRNAs that were commonly enriched in both the H3K27me3 and EZH2 immunopurified chromatin fractions were referred to as repressive chromatin associated lncRNAs. This study identified *MEG3* lncRNA as one of the top and highly conserved inactive chromatin associated lncRNAs, which was found to regulate TGF-β pathway genes through formation of RNA-DNA triplex structures. This study provided a flexibility to study lncRNAs that are functionally enriched in distinct chromatin domains as defined by signature histone modifications. In addition, using the ChRIP assay, active chromatin was purified with antibodies against H3K4me2 and WDR5. WDR5, a part of MLL1/MLL complex, catalyses the formation of H3K4me2 and H3K4me3. RNA isolated from the immunopurified chromatin fractions identified 209 chromatin associated lncRNAs [28], commonly enriched in both the H3K4me2 and WDR5 immunopurified chromatin fractions, and named these lncRNAs as active lncCARs. 43% of these active lncCARs mapped to divergent (XH) transcription units. Active XH transcription units were identified to be enriched with H3K4me2, H3K4me3 and WDR5. Active XH CARs depletion resulted in the loss of expression of the corresponding protein coding genes along with loss of H3K4me2, H3K4me3 and WDR5 at the active XH promoters. This study unravelled a new aspect of chromatin-based regulation at the divergent XH transcription units by this newly identified class of H3K4me2/WDR5 chromatin enriched lncRNAs. This approach of identifying EZH2 and WDR5 interacting lncRNAs, enriched in the H3K27me3 and H3K4me2 chromatin regions, respectively, was one of the initial approaches of mechanism-based screening for functional chromatin bound lncRNAs.

### 3.3. Profiling Interacting RNAs on Chromatin by Deep Sequencing (PIRCh-seq)

PIRCh-seq was developed to identify chromatin-associated transcriptome using antibodies recognizing histone H3, and six other distinct histone modifications associated with both active and repressive chromatin states [95]. This study additionally integrated the profiles of RNA secondary structure and RNA m6A modification to identify RNA sequences that are in contact with chromatin. Further, the authors have also characterized single nucleotide variants that define allele-specific RNA-chromatin interactions. This study has many parallels with the previous studies published using ChRIP-seq [28,33]. Both PIRCh-seq and ChRIP-seq techniques were based on the same principle i.e., antibody-based chromatin RNA immunoprecipitation. Thus, it would be interesting to compare the data obtained from these techniques to identify common chromatin associated lncRNAs which may play an important role in organizing different chromatin compartments.

### 3.4. GRID-seq

Global RNA interactions with DNA by deep sequencing (GRID-seq) comprehensively characterizes all potential chromatin associated RNAs and their cognate DNA binding regions in an unbiased fashion [96]. This exploits the principle of proximity ligation where a bivalent linker is used to ligate RNA to DNA in situ on fixed nuclei. This approach identified distinct classes of *cis*- and *trans*- acting chromatin associated RNAs that included large sets of both coding mRNAs and ncRNAs that bind to active promoters and enhancers, especially super-enhancers (Table 2).

### 3.5. MARGI-seq

Mapping RNA-genome interactions (MARGI-seq) is also based on proximity ligation where chromatin associated RNAs were ligated to target DNA using specially designed RNA and DNA linker sequences. Successfully ligated products, in the form of RNA-linker-DNA, are selected and converted to cDNA and subjected to paired-end sequencing [97]. By using human pluripotent embryonic stem cells and human embryonic kidney cell lines, MARGI-seq has identified several chromatin-associated RNAs, including well characterized lncRNAs with chromatin regulatory properties such as *Xist*, *NEAT1* and *MALAT1*. Most of the MARGI-reported lncRNA attachment regions across the genome are enriched with active histone modifications such as H3K27ac and H3K4me3.

### 3.6. ChAR-seq

This is yet another high-throughput method to characterize RNA-chromatin interactions based on proximity ligation where RNA-DNA ends were preserved in the context of chromatin followed by in situ ligation of RNA with oligonucleotide bridge containing biotin modification and DpnII restriction site. After second strand synthesis using oligonucleotide bridge, DpnII digestion and ligation reaction were performed to capture DNA-RNA contact points [98]. This method was employed in *Drosophila* cells and characterized three types of RNAs: nascent RNA transcripts close proximity to their start sites, small RNAs involved in transcription elongation and RNA processing, RNAs involved in dosage compensation.

In all of the above methodologies, RNAs that were found to be chromatin bound were mostly exonic or intronic mRNA population compared to lncRNAs/ncRNAs. In the majority of these technologies, except ChRIP-seq and PIRCh-seq, nascent RNAs are not excluded in their pulldowns. A comparative analysis of the data generated from these various approaches has been summarized and presented in Table 2.

One explanation is that all these techniques (except ActD treated ChRIP-seq) are identifying transcription dependent or transcriptionally coupled targeting of lncRNAs to the chromatin. Arguably, any positive correlation between nascent and steady state levels of an RNA and its chromatin enrichment is an indicative of transcriptional background. Although, nascent transcripts and/or the act of transcription plays an important role in gene regulation, in this review, however, we would like to focus our discussion mainly on mature functional chromatin bound lncRNAs. Moreover, each of the studies have re-validated chromatin targeting of already well characterized abundantly expressed lncRNAs, rather than identifying new targets. This argues for strategies that might help to reduce co-transcriptional purification of RNAs with chromatin.

Both RNA and non-RNA centric approaches/techniques identified genomic targets of individual lncRNAs, and also global targets of chromatin bound lncRNAs, respectively. Cumulatively, these evidences emphasize that on a global scale chromatin targeting of lncRNAs is a prominent mechanism and that these chromatin targeted lncRNAs exert their functionality mainly by fine tuning chromatin architecture resulting in an altered transcriptional readout. With the identification of such large-scale global chromatin association of several lncRNAs, the next pertinent essential question was to address the mechanism of chromatin targeting of lncRNAs.

## 4. Mechanisms by which lncRNA Targeted to Chromatin

The most persisting and pertinent question in this field has been to identify and establish unifying principles that can define and predict the functionality of lncRNAs and in a sense provide “specificity” to their modus operandi. Recent computational approaches have been trying to find signature sequence motifs in lncRNAs that can define or predict their interacting protein partners, localization and function [101,102,103,104]. Since there is no sequence conservation among the functionally conserved lncRNAs across the evolutionary spectrum, it is not advisable to look for sequence motifs that can dictate the association with DNA or transcription factor. However, by comparing the transcriptomes of 17 species, short patches of sequences were identified in lncRNAs that are evolutionary conserved and also full length orthologous lincRNA sequences from different species [102]. The latter effort has opened up a new way of looking into possible sequence information to identify conserved motifs that signify functionality. A more recent study used sequence comparison method to deconstruct linear sequence relationships in lncRNAs and evaluated similarity based on the abundance of short motifs called *k-mers*. Despite lack of sequence homology, lncRNAs with related functions had similar *k-mer* profiles, and also *k-mer* profiles correlated with protein binding and subcellular localization of lncRNAs [104]. Interestingly, a recent investigation by functionally screening the libraries of short fragments tiling across nuclear enriched lncRNAs and mRNAs, identified short sequences from Alu repeats, and C rich motifs that dictate the nuclear localization. Furthermore, the study implicated hnRNPK protein in the nuclear accumulation of lncRNA and mRNAs [103]. Certainly, the latter evidence constitutes a significant advancement in identifying RNA sequences that dictate function and localization, but we are far from understanding how lncRNAs are targeted to chromatin in a sequence specific fashion. In several contexts, the binding sites for several chromatin regulatory RNAs were characterized on a genome-scale. In addition, as explained earlier, several novel technologies were developed to characterize global RNA-DNA contacts. Surprisingly, none of these studies provided functional sequences that dictate chromatin targeting. However, global chromatin occupancy of *MEG3*, *HOTAIR* and *roX* lncRNAs revealed enrichment of GA rich sequences, which are potential landing sites for these lncRNAs. Although, common GA rich sequences were identified among the three RNA pulldowns, how three RNAs obtain specificity in targeting to their target genes is currently unknown [33,34,101]. Based on the published literature, here we summarize the mechanisms by which lncRNAs are targeted to chromatin. In principle, RNAs may be associated with chromatin via one of the three modes (Figure 2).
**Histone modifications, chromatin and DNA modifiers in the chromatin enrichment of lncRNAs:** lncRNAs, which act as a scaffold and/or guide, are targeted to chromatin by proteins having dual RNA- and DNA binding capabilities like hnRNPK [71], PGC1α [105], PRC2 [65,66], YY1 [68,69], CTCF [70], DNMTs [106]. Alternatively, lncRNAs get targeted to chromatin by interacting with RNA binding proteins (RBPs) that facilitate interaction with additional DNA binding proteins, like hnRNPU [89] (Figure 2). It is important to emphasize here that both *cis-* and *trans*-acting lncRNAs can be targeted in this way, contrary to the prevailing view that *cis* acting chromatin bound lncRNAs are mostly coupled to transcription [107,108,109,110] In contrast to the actual definition of *cis* action being “on the same chromosome”, but over time it has been erroneously conceptualized as “action restricted to site of synthesis/transcription”. The best studies of *cis* regulation of chromatin bound lncRNAs comes from classical genomic imprinting loci where imprinted lncRNAs are monoallelically transcribed and are targeted to silence multiple genes on the same chromosome as exemplified from studies of mouse and human *Kcnq1ot1* lncRNAs [29,30,32], *Airn* [42,111,112], *Xist* [42,112,113] etc. Chromatin targeting of H3K4me2 and WDR5 bound lncCARs (Active XH lncCARs) have been shown to be essential in maintaining active transcription of neighboring protein coding genes [28]. Chromatin targeting of active XH lncCARs occurs in part via WDR5 which has the potential to interact with both RNA and H3K4me2, an active histone chromatin mark. Thus, divergent transcription units enriched with H3K4me2 could recruit active XH lncCARs via WDR5. Similarly, recruitment of inactive CARs to their target genes could in part occur via EZH2, a PRC2 component with potential for the interactions with RNA and histone H3K27me3 [33].**RNA:DNA triplex:** Formation of triple helix nucleic acid structures involves Hoogsteen base-pairing interactions between RNA and the major groove of double-stranded DNA [114,115]. This RNA–DNA interaction has a stringent requirement for both polypurine sequence in DNA and a length restriction. Triplexes can form both in vitro and in vivo contexts and factors like GC content, extent of sequence complementarity, histone H3 tails, triplex target site (TTS) proximity to nucleosome entry site and open chromatin structure influence the stability of triplexes [116]. Multiple lncRNAs (having triplex forming sequence called Triplex Forming Oligonucleotides or TFOs) appear to use this mechanism to directly target specific complimentary sequences across the genome (Triplex Forming Regions or TFRs) to exert their regulatory functions (Figure 2, Table 3). Best examples of DNA:RNA triplex formation by lncRNA with specific DNA sequences include *pRNA*, which represses *in cis* the transcription of rRNA genes by targeting DNMT3b to their promoters [56], *Fendrr* which regulates developmental genes by recruiting the PRC2 complex [50], *PARTICLE* which regulates the expression of MAT2A in response to low-dose radiation [117], *MEG3* which guides PRC2 to the regulatory regions of TGF-β pathway genes [33] and *PAPAS* which guides the CHD4/NuRD complex to the rDNA promoter [118]. Recently, a global approach mapped RNA: DNA triplexes genome-wide using protein free-nucleic acids, isolated from chromatin. This approach re-validated known triplex forming lncRNAs and also identified several novel candidate lncRNAs that may execute their actions via triplex formation [119]. Besides the latter experimental approach, a computational method called Triplex Domain Finder (TDF) has been developed to detect triplex forming regions in lncRNAs, and triplex target regions across the human genome. This method successfully validated DNA-binding domains of known triplex forming lncRNAs such as *Fendrr*, *HOTAIR* and *MEG3* [101]. Two important aspects need to be considered about specificity of triplex formation mediated targeting of lncRNAs to chromatin. Firstly, there is generic sequence feature (polypurine stretch or TFOs) in lncRNAs that dictates its ability to form triplex at the genomic regions with TFRs. This still lacks one to one specificity. The question in that case remains whether any lncRNA with triplex forming capability can be targeted to all the “triplex targetable” i.e TFRs at genomic locations? and secondly, which factors initiate, promote and maintain triplex formation at target locations and that in principle is there a possibility of any difference between *cis* and *trans* targeting of triplexes lncRNAs (Figure 2)?**R-loop formation:** R-loops are three stranded RNA/DNA structures, which form co-transcriptionally at guanine-rich clusters (G-clusters) in the template strand during gene transcription [145,146]. It has been shown that RNAs containing four or more consecutive guanine residues near the 5’ end facilitates R-loop formation. R-loops, in the mammalian genome, predominantly seen at promoters and enhancers associated with GC-skewed sequences [147,148] and their formation and dynamics have been linked to transcriptional activities under physiological conditions [149,150] (Figure 2, Table 3). Recent evidence suggests that R-loop formation by lncRNAs seem to affect gene expression *in cis* through diverse mechanisms. For example, transcription of *VIM-AS1* promotes the formation of R-loop structure that was found to promote transcriptional activation of its neighboring *VIM* gene and destabilization of R-loop structure affected *VIM* expression [140]. In another context, lncRNA *GATA3-AS1* was found to be required for the formation of permissive chromatin marks H3K27 acetylation and H3K4 di/tri-methylation, at the *GATA3-AS1-GATA3* locus. Mechanistically, *GATA3-AS1* interacts with MLL1 methyltransferase and tethers to this gene locus via formation of DNA-RNA hybrid (R-loop) [151]. R-loop formation is a part of co-transcriptional process that targets nascent transcripts to chromatin *in cis*. Theoretically if any RNA with a GC-skewed sequences have the potential to form R-loop, then the pertinent question is how and in combination with which specific *in*-*cis* or *trans-* factors define the *cis-* and/or *trans-* mechanism of actions? (Figure 2).

## 5. lncRNA-Dependent Mechanisms in Chromatin Organization

First clue linking RNA and chromatin came from mammalian X chromosome inactivation wherein Xist localizes along the X chromosome undergoing inactivation. A decade after initial observation, and with the development new technologies such as chromatin immunoprecipitation (ChIP), RNA immunoprecipitation (RIP) and ChRIP, ChIRP and ChOP, we began to understand RNA-dependent chromatin changes at the onset of X inactivation, and during imprinted gene silencing. Efforts to understand common contact points along the X chromosome using ChIRP, RAP-DNA seq were not informative, rather these techniques noted that overall 3D structure allows *Xist* to spread across the inactive X chromosome (Figure 3A). Interestingly, transcription factors such as hnRNPU, YY1, with Dual RNA and DNA binding specificity, have been implicated *in-cis* function of *Xist*. Like *Xist*, *Kcnq1ot1* lncRNA has also been shown to employ 3D contacts at the genomic level in executing allele-specific gene silencing [152,153] (Table 1) (Figure 3B). However, active XH lncCARs from divergent transcription units interact with WDR5-methyl transferase complex through the RNA binding pocket of WDR5 and recruited to neighbouring protein coding gene promoters *in cis* via H3K4me2 (as WDR5 reads H3K4me2). Alternatively, active XH lncCARs directly binds to H3K4me2 enriched chromatin at the neighboring protein coding promoters and acts as a scaffold for the efficient docking of WDR5-methyl transferase complex which is necessary to maintain H3K4me2 levels and also for the conversion of H3K4me2 to H3K4me3. (Figure 3C). In this case, *in-cis* targeting of lncCARs is required to maintain transcriptional activation of neighbouring genes and does not involve spreading of the lncRNA as observed in the context of *Xist* and *Kcnq1ot1* mediated transcriptional repression (Figure 3A,B). Consistent with a role of lncRNAs in the organization of higher-ordered chromatin structure, a recent investigation implicated CTCF-RNA interaction in 3D organization of the genome wherein CTCF association with DNA is dependent on RNA and that the deletion of RNA binding zinc-finger motifs from CTCF resulted in loss of its interaction with RNA. These results convincingly document the interdependence of RNA and chromatin architectural proteins in higher-order chromatin organization. This phenomenon was also noticed in the context of eRNAs. eRNAs, bidirectionally transcribed from active enhancers, have been implicated in chromatin looping between enhancers and their cognate promoters. It is not clear whether this chromatin looping involves the interaction between enhancer binding proteins and eRNAs. However, a recent study suggested that a regulatory motif in CBP, which is enriched at the active enhancers, make specific contacts with eRNAs and this RNA interaction is required for CBP HAT activity [154]. Even lncRNAs have also been implicated in trans-chromosomal interactions as exemplified in the case of *Firre* lncRNA. *Firre* by interacting with hnRNPU, acts as a regulatory framework for promoting inter-chromosomal interactions. Considering that lncRNA is an important constituent of interphase [57] and meta-phase [155] chromatin compartments and that its functional role in gene expression by promoting intra- and inter-chromosomal interactions, we can expect to see greater insights into lncRNA mediated chromatin organization with the coevolution of technology that probe RNA-chromatin interactions.

## 6. Conclusions and Future Outlook

With the development of novel technologies to probe lncRNA-chromatin interactions and lncRNA-chromatin structure regulation, we now realize the extent of lncRNA involvement in chromatin organization. Despite improvement in our understanding of lncRNA role in chromatin-based gene regulation, biochemical and molecular details are still missing. For example, the RNA-binding specificity of PRC2 and its regulation by lncRNAs have been debated [156,157,158] due to contradictory experimental evidences. Additionally, interpretation of the role of lncRNAs in interacting or recruiting any chromatin modifiers based on existing methodologies should be concluded with more caution as exemplified by the recent observation that *HOTAIR* mediated repression is not dependent on interaction with PRC2 [159,160]. One pertinent and essential question arising from these studies is to understand the precise unifying mechanism for chromatin targeting of lncRNAs. There are two major impediments in understanding of chromatin targeting of lncRNAs. Firstly, lack of a unifying intent and approach to consolidate chromatin associated lncRNA data generated from different methodologies that is followed up by a systematic validation of interactions with novel lncRNAs rather than the same handful of well characterized imprinted lncRNAs such as *HOTAIR, MALAT1*, *NEAT1* and *MEG3* lncRNAs. Mechanistic validation of more novel chromatin bound lncRNAs would possibly lead to the identification robust unifying principles that dictate all these different modes of chromatin targeting. Secondly, barring handful of lncRNAs, chromatin interacting maps for the majority of functional lncRNAs is lacking. Motif analyses from the chromatin interacting maps of a few lncRNAs reveal enrichment of GA rich sequences. These polypurine rich sequences (TFRs) suggested to form triplexes with purine rich RNA fragments (TFOs). Two important aspects one needs to consider about the specificity of triplex formation mediated targeting of lncRNAs to chromatin. Firstly, polypurine stretch in lncRNAs dictates its ability to form triplex at genomic regions with TFRs. One major drawback with the latter suggestion is that it lacks one to one specificity. The question in that case remains whether any lncRNA with triplex forming capability can be targeted to all the “triplex targetable” i.e., TFRs at genomic locations? And secondly, which factors initiates, promotes and maintains triplex formation at target locations and that in principle is there a possibility of any difference between *cis* and *trans* targeting of triplexes lncRNAs? In addition, it would be interesting to know whether lncRNAs with potential to form triplexes have hitherto unknown conserved motifs or secondary structures that aid in targeting to chromatin. Addressing these issues would significantly enhance our understanding of mechanisms that dictate lncRNAs association with chromatin.

One of the interesting outcomes of the high-throughput approaches that were developed to probe RNA chromatin interactions was that high-amount mRNA from protein coding transcription units. It is currently not clear about the significance of chromatin association of mRNAs from protein coding genes. These mRNAs were also seen enriched in the chromatin fraction in the techniques where transcriptional inhibition was carried out using Actinomycin D. It is known that many mRNAs have noncoding variants but the proportion of enrichment of noncoding variants was not that higher compared protein coding variants. Hence it will be interesting to characterize the noncoding functions of protein coding mRNAs in chromatin organization and gene expression. Another interesting possibility might involve the role of transcription where the involved protein complexes associate in a sequence dependent (transcript *per se*) or in a sequence independent manner to target RNAs to the chromatin that facilitates an efficient regulatory condensate formation via phase separation. Both active and inactive phase separated condensates have been shown to coexist physiologically [161] and carryout gene activation or inactivation, respectively [162,163,164]. Hence it will be a very interesting possibility to explore the functional role of RNA in the formation phase separated regulatory condensate. Thus, it is now clearly evident that post high-throughput technology revolution, RNA component is taking center stage in modern biology research due to its functional versatility.

## Figures and Tables

**Figure 1 ncrna-05-00054-f001:**
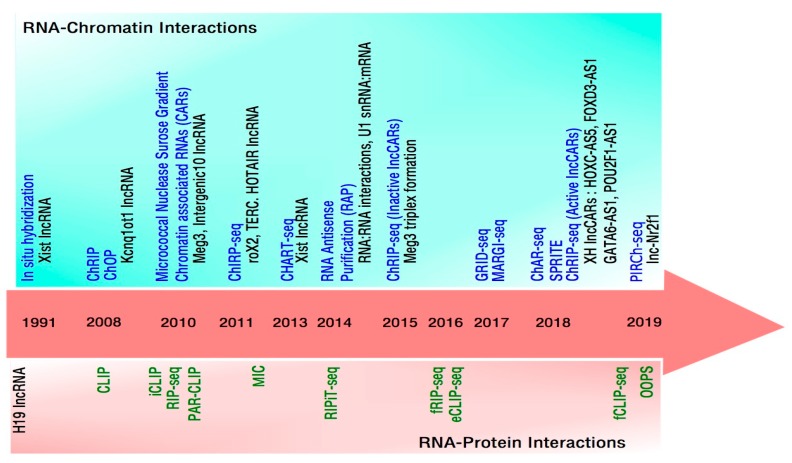
Timeline of technological advances to study RNA-Protein and RNA-Chromatin interactions. Upper panel (light sea-green box) depicts in chronological order the prominent methods (in blue) to detect RNA interactions with chromatin. Examples of some of the functionally validated lncRNAs from each of these studies are shown (in black) below the corresponding method. Middle panel depicts the year in which these methodologies were published. Lower panel (light pink box) likewise shows in chronological order methods (in green) to identify RNA interactions with proteins.

**Figure 2 ncrna-05-00054-f002:**
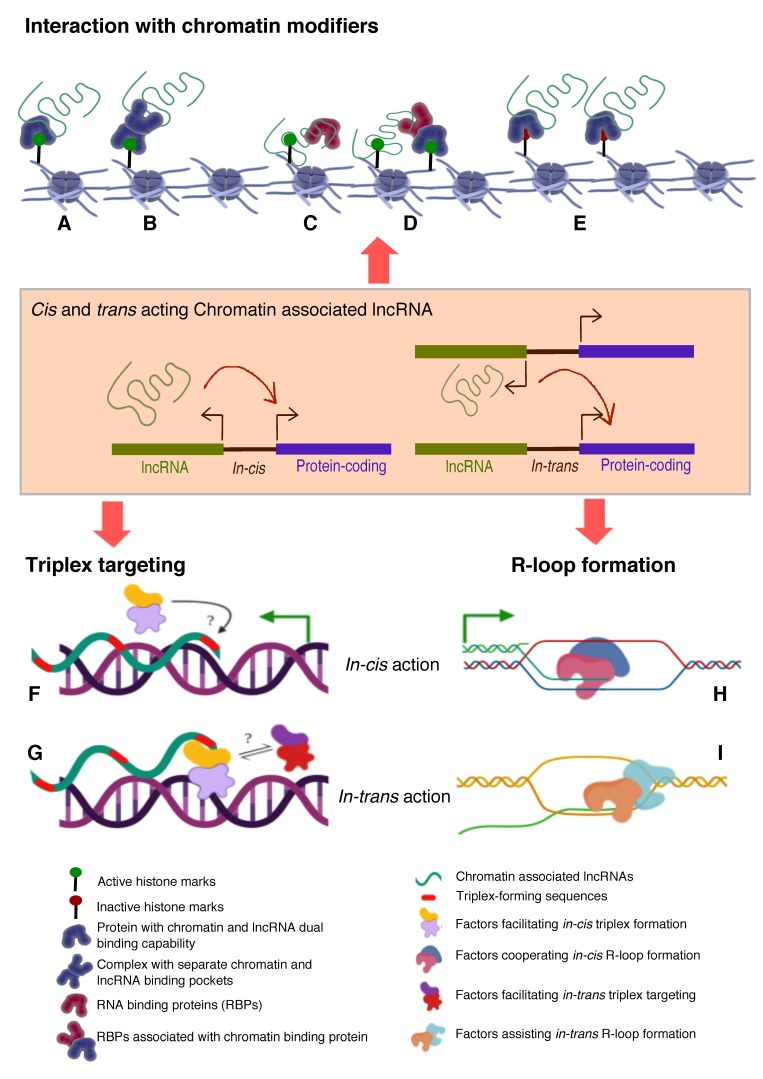
Mechanisms of chromatin targeting of lncRNAs.Three broad mechanisms that explain both *cis* and *trans* acting lncRNAs targeting to the chromatin. (**A**–**E**) depicts possible mechanisms by which lncRNAs associate with chromatin through interacting with chromatin modifiers, chromatin readers and/or RNA binding proteins. LncRNAs that interact with proteins with dual RNA-DNA binding properties can bind to chromatin enriched with active (**A**) or inactive histone modifications (**E**), or interacts with RNA binding subunit of a heterocomplex chromatin modifiers (**B**), or lncRNAs can directly be targeted (triplex or R-loop) to chromatin as a complex with any RBP (**C**) or histone modification readers can recruit RBP bound lncRNAs that can subsequently interact with chromatin via histone modifications (**D**). Inactive chromatin associated lncRNAs (iCARs) can be recruited to chromatin by a single (**E**) or heterocomplex chromatin modifiers (not shown) with histone reading as well as modifying functions and such recruitments leads to spreading of inactive chromatin through repressive histone marks. (F–G) Triplex and R-loop forming lncRNAs can target chromatin in *cis* vs. *trans* (**F**–**I**). There might be a same (**F**) or different (**G**) group of protein complexes that might play a role in either stabilizing triplex formation by *cis* (**F**) or trans-acting (**G**) lncRNAs via binding to triplex forming oligos (TFOs). Similarly, R-loop formation might be coordinated by different protein complexes *in cis* (**H**) as compared to (if any) *in-trans* targeting (**I**).

**Figure 3 ncrna-05-00054-f003:**
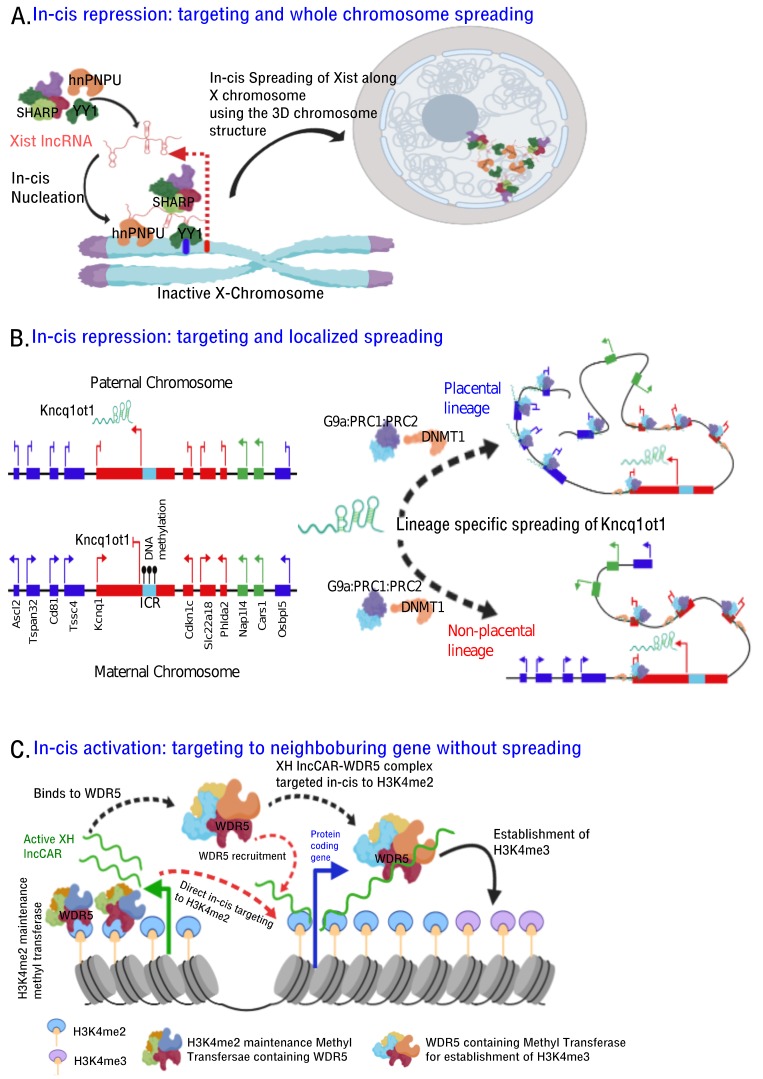
Mechanism of *in-cis* chromatin targeting. Proposed model elucidating three different mechanisms of *in-cis* chromatin targeting of some of the well characterized lncRNAs. (**A**) *Xist* lncRNA upon transcription from the X-chromosome (red bar depicts the promoter) that is due to be inactivated, interacts with YY1 protein. YY1, being bifunctional RNA-DNA interacting protein, binds to YY1 binding sites (green bar) downstream of Xist promoter thereby retaining the newly transcribed *Xist in cis*. hnRNPU is another bifunctional protein, which can interact with both chromatin and RNA, binds at the 5′ end of *Xist* and targets it to chromatin. Nucleated *Xist* lncRNA then spreads along the entire X-chromosome using the three-dimensional folding of the chromatin with the aid of other transcriptional repressor complexes such as SHARP and PRC2. (**B**) *Kcnq1ot1* lncRNA is exclusively transcribed (arrows depicting transcription) from an unmethylated paternal ICR (imprinted control region) (sky blue box), located within the intron 10 of its sense partner gene *Kcnq1* gene. It functions *in-cis* to repress (blunted arrows represent transcriptional repression) lineage specific imprinted genes. *Kcnq1ot1* (light green) interacts with and recruits G9a-PRC1-PRC2 complex to the promoters of placental linage genes (Blue boxes), while it additionally interacts with DNMT1 and targets G9a-PRC1-PRC2/DNMT1 complex to the promoters of genes that are silenced in all tissues in lineage independent fashion (Red boxes). The targeting and spreading to specific promoters across 1 mega-base region, unlike the whole X-chromosome spreading by *Xist*, is mediated by the three-dimensional folding of the chromatin. (**C**) Active XH lncCARs exemplify the case of *in-cis* targeting of lncRNAs to specific promoter regions of neighbouring protein coding genes to maintain their transcriptional activation. In the model, either the XH lncCARs first binds to WDR5-methyl transferase complex through the RNA binding pocket of WDR5 and then targeted (dashed black arrows) to chromatin at H3K4me2 (WDR5 reads H3K4me2), or they can directly bind H3K4me2 enriched chromatin (dashed red arrows) and act as a scaffold for the efficient docking of WDR5-methyl transferase complex which is necessary to maintain H3K4me2 levels and catalyse the conversion of H3K4me2 to H3K4me3. The maintenance of H3K4me2 marks is possibly mediated by a different WDR5- methyl transferase complex that is independent of the role of XH lncCARs.

**Table 1 ncrna-05-00054-t001:** Histone modifications, chromatin and DNA modifiers in characterizing chromatin associated lncRNAs: Examples of well characterized chromatin enriched lncRNAs and the techniques used to validate their detection as well as functional interaction with particular proteins.

Long Noncoding RNA	Function	Site of Action	Technique/Approach	Interacting Proteins	Ref.
**lncRNA interaction with chromatin modifying complexes**					
**Kcnq1ot1**	Lineage-specific transcriptional silencing at the imprinted *Kcnq1* locus	*In-cis*, bi-directional	Detection: Allelic RT-PCR Interaction: ChOP, ChRIP, allelic-ChIP, RIP	G9a, PRC2, DNMT1 (lineage specific interaction)	[30,32,43]
**Airn**	Lineage specific transcriptional silencing of imprinted genes *Igf2r* and *Slc22a2/3*.	*In-cis*, bi-directional	Detection: Allelic RT-PCR, RNA in-situ Interaction: RNA TRAP, RIP, allelic ChIP	*Slc22a2*/3 silencing: G9a-Airn complex *Igf2r* silencing: Transcriptional interference	[44,45]
**HOTAIR**	Transcriptional silencing of *HOXD* locus	*In-trans*	Detection: ChIP-chip of chromatin state maps (H3K4me3 and H3K27me3) in differentiating skin fibroblast cells Interaction: ChIRP-seq, Native RIP, IP	PRC2 (EZH2) LSD1 recruiting them to target gene promoters (simultaneous interaction)	[46,47]
**HOTTIP**	Homeotic gene activation at *HOXA* locus	*In-cis*	Detection: ChIP-chip of chromatin state maps in differentiating fibroblast cells, 5C Interaction: Native RIP, IP	WDR5	[48]
**Braveheart**	Activation of cardiovascular progenitor	*In-trans*	Detection: RNA-seq from mouse ESCs and differentiated tissues representing all three germ layers Interaction: In-vitro biotin-RNA pull-down, RIP	PRC2(SUZ12)	[49]
**Fendrr**	Differentiation of tissues derived from lateral mesoderm	*In-trans*	Detection: RNA-seq, ChIP-seq Interaction: RNA co-IP	EZH2, SUZ12 (PRC2), WDR5 (simultaneous interaction)	[50]
**ANRIL**	Controlling cellular senescence by transcriptional silencing	*In-cis* antisense	Detection: qPCR, RNA-FISH Interaction: CLIP	CBX7 (PRC1)	[51,52]
**Linc-Pint**	Epigenetic regulation via p53 response	*In-trans*	Detection: Custom tiling microarrays Interaction: RIP-seq	PRC2	[53]
**Chaer**	Epigenetic regulator of cardiac hypertrophy	*In-trans*	Detection: RNA-seq from pressure overload-induced mouse failing heart Interaction: RIP, Tagged RNA pull-down	EZH2 (PRC2) (66-mer motif of *Chaer* interacts in mTORC1 dependent manner)	[54]
**pRNA**	Regulation of CpG methylation at the rRNA genes	*In-cis*	Detection: RNase A treatment followed by immunofluorescence and ChIP for NoRC in NIH3T3 cells Interaction: Indirect evidence	DNMT3b recruited to DNA: RNA triplex at target promoter	[55,56]

**Table 2 ncrna-05-00054-t002:** Comparison of chromatin-associated RNA profiling methodologies: Data-based comparative analysis for the ability of these techniques to detect expression independent chromatin targeting of lncRNAs.

	ChRIP-seq (Act-D Treatment)		PIRCh-seq	GRID-seq	MARGI-seq	ChAR-seq
	Active XH lncCARs	Inactive lncCARs				
Cell lines used	BT-549	BT-549	H9, HFF, mESC, MEF and mNPC	MBA-MB-231, mESC, S2	hESC, HEK	CME-W1-cl8+
Organism	Human	Human	Human and Mouse	Human, Mouse and *Drosophila*	Human	*Drosophila*
Number of cells/chromatin required	50–60 µg chromatin per IP	50–60 µg chromatin per IP	10–20 µg chromatin per IP	5–10 µg chromatin per library	10,000–20,000 µg chromatin per library	100–400 million *Drosophila* cells per library
Crosslinking	1% Formaldehyde	UV and 1% Formaldehyde	1% Glutaraldehyde	Formaldehyde and DSG	1% Formaldehyde	1% Formaldehyde
Probes or oligos	Antibody based	Antibody based	Antibody based	Customized biotinylated bivalent linker	Ligation based: customized linker DNA.	Biotinylated oligonucleotide bridge (linker DNA)
Technical limitations	Chromatin fragment size	Chromatin fragment size	Chromatin fragment size	Frequency of AluI restriction sites in the genome	Specificity of linker ligation to RNA and the proximity of bound RNA to free DNA ends (fragment size)	Specificity of bridge ligation to RNA and the proximity of bound RNAs to free DNA ends (fragment size)
Number of chromatin bound ncRNAs	209	276	258	72 (7.36%)	Not provided	Less ncRNA and abundant mRNAs
Overrepresented class of ncRNAs	191 lncRNAs out of 209 ncRNAs	lncRNAs and novel transcripts (“cuffs”)	247 lncRNAs out of 258 ncRNAs	32 lncRNAs *MALAT1*, *NEAT1* and *U2 snRNA, roX2*, *snoRNAs*	Not provided	18% snoRNA 19% snRNA
Relation with steady state levels of nuclear expression	Chromatin enrichment of active lncCARs independent of steady state nuclear levels	Information not provided	lncRNAs overrepresented as compared to mRNAs or other ncRNAs that generally has higher expression.	Positively correlated	Positively correlated	Positively correlated
Nascent transcript enrichment	Actinomycin D treated cells were used for the assay. Functionally characterized active XH lncCARs were validated for transcription independent chromatin enrichment	Not mentioned	Less compared to GRID-seq [96], CAR [57] and CPE [99] (Chromatin pellet extract) data	Nascent transcripts are enriched	In HEK cell pxRNA peaks detected in 69.1% of all the transcription start sites. DiRNA peaks detected in 61% of all the transcription start sites	Yes. Positive correlation with (Permissive nuclear Run-On sequencing) PRO-seq data [100]
Mechanism of action	Active XH lncCARs regulate transcription *in cis* (*FOXD3-AS1, HOXC13-AS, GATA6-AS1* and *HOXC-AS2*)	One of the inactive CARs *Meg3* regulates TGF-ß pathway genes *in trans* via triplex formation	Validated chromatin targeting of *lnc-Nr2f1*	No	No	No
References	[28]	[33]	[95]	[96]	[97]	[98]

**Table 3 ncrna-05-00054-t003:** Mechanism of chromatin targeting of lncRNAs: Examples of diverse chromatin enriched lncRNAs that target chromatin via proteins with dual DNA-RNA binding properties, by triplex structure or R-loop formation. Techniques used to validate their chromatin interaction property and also their mode (triplex and R-loops) of targeting to chromatin.

Long Noncod RNA	Function	Site of Action	Technique/Approach	Interacting Protein	Ref.
**lncRNAs in interaction with RBPs with dual DNA and RNA-binding specificities**					
***linc-YY1***	YY1-mediated regulation of myogenesis	*In-trans* affects the eviction of YY1/PRC2 from the YY1 target genes	Detection: Poly A+ RNA seq from proliferating and differentiating C2C12 cells Interaction: Native IP using biotinylated in vitro-synthesized RNA. RIP suggested interaction with EZH2, SUZ12 and YY1	386–851 bp region of the *linc-YY1* interacts most efficiently with YY1.	[120]
***RMST***	*REST* dependent regulation of pluripotency and neuronal differentiation via *SOX2* pathway	*In-trans*	Detection: Custom designed micro-array. Total RNA obtained from hESCs differentiated into neural progenitors and neurons Interaction: Biotinylated antisense oligo-based RNA pull down, f-RIP	hnRNPA2/B1 and SOX2 *REST* dependent Neuronal differentiation. Dictates the binding of SOX2 at the target gene promoters, implicated in neurogenesis.	[121,122]
***LUNAR1***	Notch regulated enhancing of IGF1 signaling	*In-cis* (looping as eRNA locus)	Detection: RNA-seq generated from multiple human T-ALL cell lines and primary leukemia samples Interaction: Hi-C, ChIRP, ChIP	Interacts with IGF1R intronic enhancer to recruit Mediator and RNAP2	[123]
***linc-p21***	Transcriptional repressor in p53-dependent response	*In-trans*	Detection: RNA-seq from genetically modified cell lines for knockdown and restorable p53 levels Interaction: Biotinylated antisense oligo-based RNA pull down, f-RIP, RIP	hnRNP-K: 780 nt region at the 5′ end of lincRNA-p21interacts with hnRNPK1	[124]
***SAMMSON***	Regulation of mitochondrial homeostasis and metabolism	*In-trans*	Detection: GWAS of chromosome 3p melanoma specific focal amplification (Copy Number gain) from clinical SNP array data (TCGA) Interaction: RAP–MS, ChIRP, RAP-western blotting	p32	[125]
**lncRNAs in higher-order structures**					
***Xist***	X chromosome inactivation	*In-cis*	Detection: Allelic RT-PCR, RNA-FISH Interaction: RIP, RIP-seq, RAP, RAP–MS, ChIRP-MS	EZH2 (PRC2) SHARP, SAF-A and LBR. hnRNPK binds to a 600 nt region of *Xist* RNA and recruits Polycomb-initiating complex PCGF3/5-PRC1 *Xist* lncRNA binds 81 proteins. Protein Spen interacts via RepA region	[31,67] [58] [126] [127]
***Firre***	Role in adipogenesis; by mediating inter-chromosomal interactions	*In-cis* but colocalize with spatially proximal *trans* genomic locations	Detection: PolyA+ RNA-seq of primary brown and white adipocytes, preadipocytes, and cultured adipocytes Interaction: RAP, ChIRP, CHART, ChOP	Interacts with hnRNPU through a 156-bp repeat RNA domain	[89,90]
***NEAT1***	Nucleation and maintenance of paraspeckles	*In-trans*	Detection: qRT-PCR of HeLa cell nuclei fractionated by sucrose step-gradient centrifugation, Co-localization with paraspeckle protein PSF(SFPQ), PSP1, and p54. Custom microarray from (C2C12 cells) myoblast differentiation stages Interaction: UV-RIP, qRT-PCR of co-IP	Paraspeckle proteins	[128,129,130]
***MALAT1***	Splicing, CeRNA	*In-trans*	Detection: Subtractive hybridization in cancer cell lines Interaction: Co-RNA-FISH, qRT-PCR of co-IP	SRSF1 to regulate splicing of mRNAs	[131,132,133,134]
**LncRNAs forming R-loops and triple helixes**					
***MEG3***	Tumor suppressor lncRNA, Transcriptional repression of TGF-β pathway genes via triplex formation	*In-trans*	Detection: ChRIP-seq using antibodies against H3K27me3 and EZH2 from BT549 cell line Interaction: ChRIP, RIP, ChOP-seq, EMSA	EZH2(PRC2)	[33,57,94,101]
***Khps1***	Transcriptional activation of *SPHK1* via triplex mediated chromatin changes	*In-cis*	Detection: RNA-seq (not clear of exactly what) Interaction: RIP, Triplex forming EMSA	p300/CBP	[135,136]
***COOLAIR***	Regulates seed dormancy and flowering time through the regulation of FLC expression and flowering	*In-cis*	Detection: qRT-PCR, RNA-seq Interaction: R-loop foot printing, ChIRP	AtNDX binds to *COOLAIR* promoter to stabilize R-loops	[137,138]; [139]
***VIM-AS1***	Promote transcriptional activation of the *VIM1* gene	*In-cis*	Detection: Colon cancer hyper methylation associated positive correlation with divergent VIM1 gene Interaction: R-loop foot printing, EMSA, RNase H1 assay, DRIP with S9.6	Interacts with single stranded DNA(R-loop) to enhance NF-κB binding at the VIM1 promoter	[140]
***TERRA***	Maintenance of short telomeric structure by regulating the rate of replicative senescence		Detection: qRT-PCR after releasing G1-arrested cells into the cell cycle Interaction: DRIP with S9.6, EMSA	Interacts with telomeric DNA forming R-loops that promotes homology directed repair at very short telomeres by excluding Rif2 mediated RNase H2 recruitment	[141,142,143]; [144]

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
