# Peer review of "Understanding Long Noncoding RNA and Chromatin Interactions: What We Know So Far"

_ncrna, 2019, doi:10.3390/ncrna5040054_

Round 1

Reviewer 1 Report

In this review, Mishra and Kanduri provide a functional and technical overview of chromatin-associated long noncoding RNAs. Given recently-(re)gained interest in nuclear organisation, chromatin-associated RNA becomes an attractive topic and there certainly a room for this type of review. The authors are doing good work in reviewing different approaches that were previously utilised in order to identify and map chromatin-associated transcripts. They also describe specific examples for lncRNAs that were studied in this context, while doing well in acknowledging knowledge gaps, mainly when it comes to functional consequences of previous observations.

The authors made a significant contribution to this field in the past and this review will be of a high value for researchers entering into this field today. While there are many lncRNA-focused reviews, the focus here was made on chromatin-associated lncRNAs and thus this review is relatively unique, pointed and valuable. While the review is clear and with overall good quality, there are several specific points that should be addressed before publishing.

Major points

Within the text, Box 2 and Fig 1, the authors discussing next-generation sequencing-based approaches for the study of RNA-protein interactions. Yet, there is no discussion into proteomics-based approaches. Methods as RIC [PMID: 22658674], OOPS [PMID: 30607034], RBDmap [PMID: 29095441] and targeted RBR-ID [PMID: 30833789], among other methods, are being utilised to identify protein-RNA interactions. Even if the authors intended to write an RNA-centric review, a discussion into the study of RNA-protein interactions is incomplete without minimal coverage of proteomics-based approaches, especially as these become popular and allows for unbiased discovery of RNA-binding proteins and, in some cases, mapping RNA-binding domains.

Section 2 and Table 2 provides a very useful description of methods for the discovery and characterisation of chromatin-associated RNA. But it would be good to include some practical guide for method selection, including some technical specifications, as the number of cells needed, if/what modified oligonucleotides are required, how sensitive the method is (i.e. how many lncRNA molecules-per-cell are required for detection, based on previous works), if crosslinking required, or any other information that could guide method selection. For instance, this can be done by adding rows to Table 2, or through a separate table of Figure.

Most of the lncRNA in Table 1 (6/10) binds to PRC2. The authors should minimally mention somewhere in the text that the RNA-binding specificity of PRC2 and its regulation by lncRNAs have been subjected to various views and has not yet fully understood mechanistically (for reviews focused on this topic, see Brockdorff 2013 [PMID: 23431328]; Davidovich and Cech 2015 [PMID: 26574518]; Ringrose 2017 [PMID: 28934594])

Minor points

The authors use the terms “covalent modifications” of histones when talking about histone modifications (e.g. see section 4.1). The authors also refer to “covalent interactions” when talking about interactions between RNA to proteins or proteins to chromatin (e.g. see Fig 2). The usage of "covalent" in this context is confusing, as covalent bonds are formed when electron pairs are shared between atoms. Accordingly, covalent bonds are present in all histone modifications, but not when a protein binds to RNA or chromatin through hydrogen bonds, Van der Waals and/or electrostatic interactions. Hence, are there any histone modifications that do not involve covalent bonds? How common are protein-RNA or protein-chromatin interactions that do involve covalent bonds? In order to avoid these questions, I would consider revising the text and figures to remove 'covalent' when it is redundant with 'modifications' or simply wrong.

L114-5: The definition of lncRNA typically include both the length, as well-stated by the authors, but also the lack of ORF above a certain length (typically 30 amino acids). Furthermore, a citation is needed at the end of the sentence defines lncRNA.

L457-9: The sentence starts with “Positive correlation…” is complex in nature and the lack of punctuations, which makes it unclear. Consider revising.

Shouldn’t PAN RNAs be mentioned somewhere in this review? They were also reported to be associated with chromatin (e.g. PMID: 30383841).

Author Response

In this review, Mishra and Kanduri provide a functional and technical overview of chromatin-associated long noncoding RNAs. Given recently-(re)gained interest in nuclear organisation, chromatin-associated RNA becomes an attractive topic and there certainly a room for this type of review. The authors are doing good work in reviewing different approaches that were previously utilised in order to identify and map chromatin-associated transcripts. They also describe specific examples for lncRNAs that were studied in this context, while doing well in acknowledging knowledge gaps, mainly when it comes to functional consequences of previous observations.

The authors made a significant contribution to this field in the past and this review will be of a high value for researchers entering into this field today. While there are many lncRNA-focused reviews, the focus here was made on chromatin-associated lncRNAs and thus this review is relatively unique, pointed and valuable. While the review is clear and with overall good quality, there are several specific points that should be addressed before publishing.

Major points

Within the text, Box 2 and Fig 1, the authors discussing next-generation sequencing-based approaches for the study of RNA-protein interactions. Yet, there is no discussion into proteomics-based approaches. Methods as RIC [PMID: 22658674], OOPS [PMID: 30607034], RBDmap [PMID: 29095441] and targeted RBR-ID [PMID: 30833789], among other methods, are being utilised to identify protein-RNA interactions. Even if the authors intended to write an RNA-centric review, a discussion into the study of RNA-protein interactions is incomplete without minimal coverage of proteomics-based approaches, especially as these become popular and allows for unbiased discovery of RNA-binding proteins and, in some cases, mapping RNA-binding domains.

A) Thanks for your suggestion. We have added RIC, RBBmap and OOPS in Box 2. We also included RIC and RBBmap in the Fig.1.

Section 2 and Table 2 provides a very useful description of methods for the discovery and characterisation of chromatin-associated RNA. But it would be good to include some practical guide for method selection, including some technical specifications, as the number of cells needed, if/what modified oligonucleotides are required, how sensitive the method is (i.e. how many lncRNA molecules-per-cell are required for detection, based on previous works), if crosslinking required, or any other information that could guide method selection. For instance, this can be done by adding rows to Table 2, or through a separate table of Figure.

A)  Thanks for your suggestions. We have added some of the technical aspects to Table 2 as suggested.

Most of the lncRNA in Table 1 (6/10) binds to PRC2. The authors should minimally mention somewhere in the text that the RNA-binding specificity of PRC2 and its regulation by lncRNAs have been subjected to various views and has not yet fully understood mechanistically (for reviews focused on this topic, see Brockdorff 2013 [PMID: 23431328]; Davidovich and Cech 2015 [PMID: 26574518]; Ringrose 2017 [PMID: 28934594])

A)  We have addressed this point in Conclusion section

Minor points

The authors use the terms “covalent modifications” of histones when talking about histone modifications (e.g. see section 4.1). The authors also refer to “covalent interactions” when talking about interactions between RNA to proteins or proteins to chromatin (e.g. see Fig 2). The usage of "covalent" in this context is confusing, as covalent bonds are formed when electron pairs are shared between atoms. Accordingly, covalent bonds are present in all histone modifications, but not when a protein binds to RNA or chromatin through hydrogen bonds, Van der Waals and/or electrostatic interactions. Hence, are there any histone modifications that do not involve covalent bonds? How common are protein-RNA or protein-chromatin interactions that do involve covalent bonds? In order to avoid these questions, I would consider revising the text and figures to remove 'covalent' when it is redundant with 'modifications' or simply wrong.

A) Thanks for your suggestion. We have removed the term covalent.

L114-5: The definition of lncRNA typically include both the length, as well-stated by the authors, but also the lack of ORF above a certain length (typically 30 amino acids). Furthermore, a citation is needed at the end of the sentence defines lncRNA.

A) We have revised the text and added appropriate citation.

L457-9: The sentence starts with “Positive correlation…” is complex in nature and the lack of punctuations, which makes it unclear. Consider revising.

A) Thanks for your suggestions. We have revised the sentence

Shouldn’t PAN RNAs be mentioned somewhere in this review? They were also reported to be associated with chromatin (e.g. PMID: 30383841). 

A) We have decided not to include this study in our review.

Reviewer 2 Report

The review is very long and tries to cover everything we know about non-coding RNA, methods, history and significant findings. That results in a confusing paper that does not give a good background into any of the topics.

Methods described are general for RNA work and it is far too extensive for a single review paper to describe them all in a meaningful way.

The review should be focused (on findings, methods or history), shortened and language improved to make up an interesting and useful review.

Author Response

The review is very long and tries to cover everything we know about non-coding RNA, methods, history and significant findings. That results in a confusing paper that does not give a good background into any of the topics.

Methods described are general for RNA work and it is far too extensive for a single review paper to describe them all in a meaningful way.

The review should be focused (on findings, methods or history), shortened and language improved to make up an interesting and useful review.

A). This review is primarily focused on understanding the interaction between RNA and chromatin and we reviewed the literature accordingly. We have now removed BOX1 which covered general RNA-sequencing technologies and retained BOX2 (now BOX1) which is important for the review. We have gone through the review carefully and corrected for the language mistakes.

Reviewer 3 Report

This review is focused on the noncoding RNA associated to chromatin that is involved in chromatin structure regulation defining transcriptional readout. It includes short   description of technologies attributed to detection of chromatin associated lncRNA which are supported with necessary references being very useful to readers. The possible regulatory function of lnsRNAs in R-loop and triplex formation is considered in detail, with an abundance of cited papers.
This review undoubtedly deserves publication, representing a professional analysis of the amount of work on non-coding RNA in chromatin.

Comment

The characteristics of lncHOTAIR in table 1 should be changed with relevant  comments in the text, taking into account the work of Portoso M et al 2017 that showed the dispensability of HOTAIR's participation in PRC2 repression. The answer of Blanco MR and Guttman M paper to Portoso`s publication concerns a possibility of artifact obtaining using the RIP and CLIP methods without denaturation. These results case in the course of Hotair studies I guess well be useful to mention when characterizing these methods. The same goes in caution in interpreting the experiments reported by Kaneko et al  (p.11, lines 276-278) and Chu 2011 (line 318).

Minor comments

Some references are needed to be verified (lines 522..529 ,551, 640)

 Line 521 -522 - lncRNAs are targeted to chromatin by COVALENTLY? interacting with proteins.

Author Response

This review is focused on the noncoding RNA associated to chromatin that is involved in chromatin structure regulation defining transcriptional readout. It includes short   description of technologies attributed to detection of chromatin associated lncRNA which are supported with necessary references being very useful to readers. The possible regulatory function of lnsRNAs in R-loop and triplex formation is considered in detail, with an abundance of cited papers. 
This review undoubtedly deserves publication, representing a professional analysis of the amount of work on non-coding RNA in chromatin.

Comment

The characteristics of lncHOTAIR in table 1 should be changed with relevant  comments in the text, taking into account the work of Portoso M et al 2017 that showed the dispensability of HOTAIR's participation in PRC2 repression. The answer of Blanco MR and Guttman M paper to Portoso`s publication concerns a possibility of artifact obtaining using the RIP and CLIP methods without denaturation. These results case in the course of Hotair studies I guess well be useful to mention when characterizing these methods. The same goes in caution in interpreting the experiments reported by Kaneko et al  (p.11, lines 276-278) and Chu 2011 (line 318).

A) Thanks for your suggestion. We have addressed this point in conclusion section

Minor comments

Some references are needed to be verified (lines 522..529 ,551, 640)

A)

Line 522-529: equivalent to present 431: references have been added as suggested

Line 551: present line 455: references added.

Line 640: present line 546: Checked the references, looks fine. 

 Line 521 -522 - lncRNAs are targeted to chromatin by COVALENTLY? interacting with proteins.

A) Thanks for noticing this mistake and we have removed the term covalent.

Round 2

Reviewer 2 Report

The review gives a good overview of ncRNA work but I find too long and not sufficiently focused to grasp the attention of an interested reader.